# Direct Simulation Monte Carlo Simulation of the Effect of Needle Valve Structures on the Rarefied Flow of Cold Gas Thrusters

**DOI:** 10.3390/mi14081585

**Published:** 2023-08-11

**Authors:** Songcai Lu, Xuhui Liu, Xudong Wang, Shurui Zhang, Yusong Yu, Yong Li

**Affiliations:** 1Beijing Institute of Control Engineering, Beijing 100190, China; songcai@petalmail.com (S.L.); xhliu99@163.com (X.L.); kyffwd@163.com (X.W.); 2Hydrogen Energy and Space Propulsion Laboratory (HESPL), School of Mechanical, Electronic and Control Engineering, Beijing Jiaotong University, Beijing 100044, China; 21121387@bjtu.edu.cn (S.Z.); ysyu@bjtu.edu.cn (Y.Y.)

**Keywords:** cold gas micro-nozzle, rarefied flow, DSMC method, needle valve opening ratio, large length-to-diameter ratio, micro-channel

## Abstract

The needle valve, serving as the flow control unit of the thruster system, is a crucial component of the entire thruster. Its performance directly impacts the flow state of the rarefied gas in the micro-nozzle structure of the cold gas micro-thruster, thereby exerting a significant influence on the high precision and stability of the propulsion system as a whole. This study examines the impact of different needle valve structures on the flow and thrust in micro-nozzles using the DSMC method. The analysis includes discussions on the spatial distribution, Kn distribution, slip velocity distribution, and pressure distribution of the micro-nozzle’s flow mechanism. Notably, increased curvature of the needle valve enhances the flow velocity in the throat and expansion section. The magnitude of the curvature directly affects the flow velocity, with larger curvatures resulting in higher velocities. Comparing different spool shapes, the conical spool shape minimizes the velocity gradient in the high-speed region at the junction between the spool area and the outlet pipe, particularly with a wide opening. Increasing the curvature of the spool leads to a higher velocity in the expansion section. Consequently, an arc-shaped spool valve maximizes the nitrogen flow at the nozzle during wide openings, thereby enhancing thrust. These research findings serve as a valuable reference for the structural design of the needle valve in the micro-nozzle of the cold gas micro-thruster.

## 1. Introduction

In recent years, various applications such as space gravitational wave detection and high-precision earth gravity field measurement have created a pressing demand for micro-Newton thrusters that offer a wide adjustment range, high accuracy, and exceptional resolution [1,2,3]. Achieving precise measurements of fundamental physical data in space necessitates reducing the residual disturbance acceleration noise of the satellite platform within the measurement frequency band to below 10^−15^ m/s^2^/Hz^1/2^ [3]. Consequently, it is crucial to equip satellite platforms with micro-Newton thrust capabilities to counteract the influence of non-conservative forces, including atmospheric effects, solar light pressure, and cosmic particles. This setup establishes a flight environment dominated by “pure gravity”, enabling drag-free control and ensuring that the payload operates in an ultra-stable working environment. Currently, several prominent micro-Newton thruster technologies exist worldwide, including cold gas thrusters, electrospray thrusters, RF ion thrusters, and cusped field thrusters [4,5,6]. Among these, cold gas thrust technology stands out for its high reliability and capacity to achieve a wide range of thrust adjustments, spanning from 0.15 to 1000 μN, making it a significant focus of international development [7,8,9].

The micro-nozzle serves as a critical component of the cold gas micro-thruster. Investigating the gas flow mechanism within the micro-nozzle forms the foundation for achieving wide-ranging variable thrust control at the micro-Newton level [10]. In the cold gas micro-thruster, the micro-nozzle is controlled by a piezoelectric actuator, allowing for the precise adjustment of the throat’s flow area. Propellant expands through the nozzle and is expelled to generate thrust. The average free path (λ) of the gas working medium at the upstream inlet of the nozzle measures approximately 0.05 μm, but it can extend to several meters at the nozzle outlet. Consequently, the flow transitions from a continuous state upstream of the nozzle to a free molecular flow state near the nozzle outlet, covering the entire spectrum. Therefore, gaining an in-depth understanding of the gas flow characteristics across the spectrum and scales within the cold gas thruster is vital for achieving precise thrust control [11,12].

Figure 1 illustrates the structure of a typical cold gas micro-nozzle, which consists of four main parts: the needle valve, pressure accumulator, microchannel, and expansion segment. For this study, the inlet flow pressure ranges from 0.1 to 0.25 MPa, the microfluidic channel diameter is 0.3 mm, and the expansion nozzle outlet diameter measures 5 mm. The needle valve, functioning as the flow control unit of the thruster system, plays a pivotal role in the overall performance, precision, and stability of the propulsion system. Firstly, the needle’s shape determines the flow rate, also known as its throttling performance. Secondly, due to changes in the flow area, the needle’s pressure and velocity undergo sudden variations, especially at smaller openings where high-speed flow flushing poses an increased risk of damage. Moreover, the needle’s shape affects the flow within the channel, influencing flow stability and generating thrust noise. Consequently, the needle’s shape significantly impacts the gas flow characteristics inside the thruster, consequently affecting the operational performance and service life of the cold gas thrust system. Given the inherent challenges in experimental measurements due to the extremely low thrust produced by micro-propulsion, numerical simulation methods are of great significance for simulating the flow characteristics of micro-nozzles, facilitating the design and improvement of micro-space propulsion systems [13,14].

The micro-nozzle’s thrust performance is significantly affected by the nozzle structure and incoming flow conditions. In recent years, the direct simulation Monte Carlo (DSMC) method has been largely used to predict the flow field inside micro-channels or micro-nozzles [15,16]. Darbandi and Roohi [14] conducted computational analysis of the flow Kn number inside a micro-jet nozzle. The flow inside the micro-jet nozzle exhibits a complex distribution of Kn numbers, with high Kn numbers predominantly concentrated in the peripheral region downstream of the nozzle throat expansion section. Furthermore, the expansion of Mach numbers primarily occurs in the diverging section and buffer zone. Torre et al. [17] utilized an NS/DSMC hybrid numerical method to analyze the rarefied flow within micro-newton thrusters. The results indicate that friction losses and rarified effects within the micro-jets contribute to a decrease in thrust output, reducing the thrust-to-weight ratio by approximately 35% compared to the one-dimensional isentropic theory solution. Alexeenko et al. [18] conducted a performance analysis of micro-thrusters based on thermo-fluid coupling modeling and simulation and numerically simulated gas flow rate inside the micro-thruster using DSMC method. Through considering Reynolds number, thermal boundary conditions, and micro-nozzle height in detail, it was found that the thrust and mass flow rate coefficient of the three-dimensional micro-thruster decrease with time under different flow conditions. Sukesan and Shine [19] conducted the N–S and DSMC simulations for micro-nozzle flows. The rarefaction effects in the micro-nozzle with different nozzle geometry parameters were discussed. Results show an optimum divergence angle and length, maximizing the performance for a specific operating condition and nozzle size. So far, the influence of the nozzle geometry on the internal flow characteristics is still unclear. In this study, the DSMC (direct simulation Monte Carlo) method is employed to simulate the multi-scale rarefied flow characteristics within the micro-nozzle of the cold gas micro-thruster. Through altering the micro-nozzle structure and the needle valve opening, the distribution of the flow field and thrust results for multi-scale flow within the micro-nozzle under different parameters are obtained.

## 2. Numerical Simulation

### 2.1. Direct Simulation Monte Carlo Method

It is worth noting that under the specific structure and conditions of the micro-thruster examined in this study, the Knudsen number (Kn), calculated based on the inner diameter of the microfluidic channel, exceeds 0.01 when the thrust range falls below 10 μN, indicating that the gas flow within the micro-nozzle deviates from the continuum hypothesis [20]. When the continuum hypothesis is no longer applicable, phenomena such as flow shear and heat transfer cannot be adequately addressed using continuum methods. As a result, the molecular dynamics (MD) and DSMC methods were proposed to solve the Boltzmann equation and describe rarefied flow problems [21,22,23]. However, due to the MD method’s direct solution of the motion and collision of individual molecules, the computational time required is substantial. For instance, solving the second-level process of 1 cubic millimeter of air under standard conditions using the MD method may take up to 100 machine hours. Consequently, when it comes to solving rarefied flow problems at larger scales, the MD method becomes computationally demanding.

The basic equation describing the flow of rarefied flow is the Boltzmann Equation [24]:(1)$$\frac{df}{dt}=\frac{\partial f}{\partial t}+c\frac{\partial f}{\partial x}+\frac{\vec{F}}{m}\frac{\partial f}{\partial c}=\int \left(f^{*}f_{1}^{*}-ff_{1}\right)gbdbd\varepsilon dc_{1}$$
where f(t,x,c) is the motion distribution function, *t* is time, x is the molecular position vector, c and $c_{1}$ is the initial velocity before two molecules collide. Additionally, $g=\left|c_{1}-c\right|$, $f_{1}=f_{1}\left(t,x,c_{1}\right)$, $f^{*}=f^{*}\left(t,x,c^{*}\right)$, $f_{1}^{*}=f_{1}^{*}\left(t,x,c_{1}^{*}\right)$, and $c^{*}$ and $\;c_{1}^{*}$ are the speeds after the two collisions, respectively.

Bird introduced the direct simulation Monte Carlo (DSMC) method to solve the Boltzmann equation for rarefied flow, and in 1992, the consistency between the DSMC method and the Boltzmann equation was demonstrated [25]. The DSMC method serves as a simplified solution approach for the Boltzmann equation, incorporating a specific number of sample points to simulate molecular or atomic clusters for statistical sampling [26]. Furthermore, the DSMC method utilizes scattering rate and post-collision velocity distribution models to describe collision events among molecules or at the boundary between molecules and walls. Currently, the DSMC method has been used to simulate rarefied and multiscale gas flows, including those related to space spacecraft in orbit flight and microscale rarefied flow [27]. 

Numerous studies have shown that the DSMC method displays high numerical accuracy in calculating the rarefied flow in micro-nozzles [15,28,29]. The DSMC simulation method involves addressing two key challenges: collision simulation between molecules and collision simulation between molecules and walls. The details of how collisions are calculated in DSMC depend on the molecular interaction model, e.g., the hard spheres (HS) model and variable-hard-sphere (VHS) model [30]. The VHS model is used in this study due to its computational efficiency and capability to accurately describe the collision between nitrogen molecules. The effective collision cross-section of VHS model is a function of the relative velocity. It allows for a more realistic representation of particle interactions in certain systems than the HS model [31]. In the rarefied flow problem of microfluidic channels, collisions between molecules and walls will affect the movement behavior of molecules in microfluidic channels [13]. Especially when the length and diameter of the microfluidic channel are relatively large, the influence of molecular collision with the wall will be more significant [13,32,33]. So far, different schemes for performing collision rates have been developed in the DSMC method. The commonly accepted and widely used no-time-counter (NTC) scheme is analyzed and discussed widely [34]. The first Bernoulli trials scheme (BT) was put forward by Belotserkovskii and Yanitskiy [35]. The BT model defines a collision probability function for each particle pair and can avoid the possibility of repeated collisions. Stefanov et al. introduced a “simplified” alternative of the Bernoulli trials scheme called ‘SBT’ [36], with a linear dependency of the computational cost on the number of particles per cell. The SBT scheme avoids repeated collision in cells and permits simulations using a much smaller mean number of particles per cell. Nevertheless, the computational costs for a larger number of particles per cell are yet significant, and the numerical efficiency of SBT is still less than the corresponding characteristics of the standard NTC collision scheme. Roohi et al. [37] proposed a generalized form of the Bernoulli trial (GBT) collision scheme to reduce the computational effort of the SBT collision model when the number of particles is comparable with that used in simulations with the NTC scheme. Taheri et al. [38] developed the symmetrized and simplified Bernoulli trials (SSBT), making the algorithm symmetric with respect to choosing the second particle. The SSBT scheme improves the quality of the selection process to achieve the same convergence limit as that of the SBT and NN schemes. The hybrid algorithm employs NTC, GBT, or SBT depending on the instantaneous number of particles in the considered cell. The novel hybrid TAS algorithm benefits from both the hybrid collision approach and the transient adaptive subcell grid covering each collision cell to achieve a uniform accuracy of order O (Δt, Δr) independently of the number of particles in the cells [39].

In this study, the DSMC solver, dsmcFoam, in OpenFOAM-9 is used. dsmcFoam is a DSMC solver for rarefied gas dynamics, implemented within the OpenFOAM software framework, and parallelised with MPI [40]. The nitrogen molecule is considered in the simulation [41]. The N_2_ was treated as polyatomic gas. The dsmcFoam solver determines intermolecular collisions for polyatomic species using the variable hard sphere (VHS) model. The phenomenological Larsen–Borgnakke model was used to distribute post-collision energy between the translational and rotational modes.

### 2.2. Simulation Models

For the inlet boundary, i.e., the left side of the calculation domain in Figure 2, set the molecular number density to 1.9011 × 10^25^ 1/m^3^ to correspond to the actual inlet pressure of 0.15 MPa. The given information states that the mass of a nitrogen molecule is 46.5 × 10^−27^ kg. Additionally, it states that the equivalent diameter of a nitrogen molecule is 4.17 × 10^−10^ m, which represents the size of the molecule. Furthermore, it mentions that the actual outlet pressure is a vacuum. This means that at the outlet surface, there is no pressure exerted by any external gases or substances. As a result, the molecular number density at the outlet surface is zero. Moreover, the wall is set to an isothermal boundary of 300 K. This thermal condition ensures that there are no temperature gradients or variations across the surface of the wall. 

Figure 2 illustrates the computational domain of the micro-nozzle, which can be divided into four sections: the inlet section, throat section, microfluidic section, and expansion section [42]. As the nozzle possesses axisymmetric properties, the model’s computational domain can be simplified to half. The key structural parameters of the micro-nozzle are as follows: (1) The cone angle of the valve needle is 30°. (2) The micro-channel has an inner diameter of 0.3 mm and a length of 3 mm. (3) The expansion section measures 3.8 mm in length, with an outlet section diameter of 5 mm. 

Table 1 presents the number of axial grids for each region in sequential order. The impact of the mesh quantity on the calculation results is analyzed in the same table. During the grid test, the inlet molecular number density remained constant at 1.9011 × 10^25^ 1/m^3^, the time step was set to 1 × 10^−9^ s, and the needle valve opening was 10%. Based on the comparison results of the average velocity at the outlet, it is evident that Grid3 exhibits high computational accuracy with a relatively small total number of meshes. Furthermore, the effect of further increasing the mesh on the relative deviation of the calculation results is found to be less than 2%.

The effect of the molecular number density and average particle number per cell on the calculated average nitrogen gas speed at the outlet is shown in Figure 3. According to the comparison results of the average gas speed at the outlet, the molecular number density of 5.5 × 10^8^, i.e., the average particle number per cell of 65, provides high numerical accuracy. The relative deviation of the calculation results caused by continuing to reduce the equivalent molecular number is 1.06%. When the average particle number per cell falls below 65, the computed relative error increases rapidly. Therefore, the molecular number density and average particle number per cell are set to 5.5 × 10^8^ and 65, respectively. A fixed calculation time step is used, which takes the value of 1 × 10^−9^ s. The flow state in the micro nozzle reaches a quasi-stable state at 0.1 ms [41].

Two types of spool shape models have been developed using common spool forms. The purpose is to assess the impact of spool shape on the flow characteristics of needle valves within the cold gas propulsion system. The two types, concave and convex, were chosen, distinguished by their varying curvatures. Considering the geometric characteristics and practical processing considerations, four styles with curvatures of ±277.8 1/m and ±555.6 1/m were ultimately selected, as illustrated in Figure 4.

### 2.3. Validations of the Numerical Results

To validate the accuracy of the simulation, thrust data at different flow rates (adjusted through controlling the needle valve opening) were calculated and compared with experimental data obtained under identical boundary conditions. The test setup and test methods are described in reference [41]. The experiment was conducted with an inlet pressure of 0.15 MPa, working medium temperature of 293.15 K, wall temperature of 293.15 K, and vacuum outlet. The results, as depicted in the figure, show that the thrust exhibits a similar trend with the opening of the needle valve, and the magnitude of thrust is comparable to the experimental results, demonstrating good agreement (Figure 5). It is important to note that accurately calculating low needle valve openings requires precise modeling and accounts for factors such as molecular interactions and molecular vibrations. Additionally, the measurement of thrust at low needle valve openings poses challenges and introduces errors.

## 3. Results and Discussion

### 3.1. Throttling Characteristic

The needle valve plays a crucial role in regulating the flow, making flow characteristics highly significant. This study compares the change in nozzle outlet flow rate under different spool shapes and varying degrees of valve opening. As shown in Figure 6, it is evident that the mass flow rate at the outlet exhibits a linear improvement with increasing openness. This is since the throat area is approximately proportional to the valve displacement, while the mass flow rate is proportional to the throat area. Comparing the curves with different curvatures, it is observed that changing the curvature leads to an enhancement in the mass flow rate at the outlet. In terms of structural influence, under concave or convex conditions, a greater curvature results in a higher mass flow rate at the same opening degree. In particular, when the curvature is positive and the needle valve is convex, the increase in mass flow rate is more pronounced when comparing the same absolute value of curvature.

The mass flow rate at the outlet for different curvatures and openings is compared to the cone, and the corresponding percentages are presented in Table 2. The results reveal consistent trends in the influence of curvature on the mass flow rate across different openings. In both concave and convex configurations, an increase in the absolute curvature value corresponds to an increase in the mass flow rate. Moreover, the effect is more pronounced when the curvature is positive. These findings indicate that the curvature of the needle valve significantly impacts the mass flow rate, with a greater curvature leading to a higher flow rate. This information emphasizes the importance of considering curvature as a crucial factor in optimizing and controlling the mass flow characteristics of the system.

Figure 7 illustrates the pressure distribution in each area of the nozzle at a 50% opening, specifically at a time of 1 millisecond. The left panel of the figure presents the results obtained for different curvatures under the condition of a 50% opening of the needle valve. On the other hand, the right panel showcases the pressure variation curve along the axis with the length-diameter ratio for different curvatures. The pressure gradually diminishes from the inlet to the outlet as the high-pressure gas flows out from the inlet, as observed in the figure. Altering the curvature can moderate the change in pressure gradient at the throat, facilitating the entry of high-pressure gas into the microchannel. Moreover, an increase in curvature causes the high-pressure region to advance downstream within the microchannel. This phenomenon arises due to the convex shape of the needle valve, allowing more space for the gas in the region before entering the microchannel. Similarly, the concave needle valve promotes smoother gas flow into the microchannel, exhibiting favorable effects in terms of high-pressure expansion. These findings highlight the significance of needle valve curvature in controlling pressure distribution and optimizing the performance of the nozzle. Through carefully adjusting the curvature, it is possible to manipulate the pressure gradient and enhance the efficiency of gas flow in the microchannel, ultimately improving the overall functioning of the propulsion system.

Needle valves with different spool cone curvatures exhibit consistent pressure changes along the axis. The variations in curvature result in different flow areas, which subsequently lead to varying concentrations of nitrogen flowing through the valve. Consequently, different pressures are observed. In the case of a curved poppet valve core, the larger flow area reduces the accumulation of nitrogen in the front of the valve core, resulting in lower pressure. As the gas flows through the microchannel, a significant pressure drop occurs. This can be attributed to the smaller gap between the valve body and the valve core facilitated by the arc cone valve core. When the opening is larger, the changes in pressure become more gradual, leading to a lower hydrogen flow rate and higher pressure. As a result, the pressure drop is reduced. The consistent pressure changes observed along the axis with different curvatures highlight the impact of spool cone curvature on flow dynamics and pressure distribution. Through carefully selecting and designing the curvature of the needle valve, it is possible to optimize gas flow, minimize pressure variations, and improve the overall performance of the propulsion system.

### 3.2. Flow Regime Spatial Distributions and Local Kn Distribution

To depict the highly rarefied flow conditions within the micro-nozzle, Figure 8 presents the flow mechanisms in each region of the micro-nozzle and the distribution of local Knudsen numbers at 1 ms. The figure includes the distribution results for different curvatures with a 10% opening of the needle valve, the variation curve of local Knudsen numbers along the axis with the length–diameter ratio for different curvatures, and the proportion of each region along the axis for different curvatures. The color bar distribution adopts a logarithmic scale to visualize the positions of Knudsen numbers 0.01, 0.1, and 10. The results demonstrate that when the needle valve opening is set at 10%, the contour surface with a Knudsen number of 0.01 appears in the vicinity of the throat, while the contour surface with a Knudsen number of 0.1 is close to the downstream expansion section of the microchannel. The region between these two contour surfaces represents the slip flow region, while the yellow region at the end of the microchannel and in the nozzle extension section corresponds to the transition flow region. The red region in the nozzle extension denotes the free molecular flow region. A comparison of the images with different curvatures reveals that the position of the contour surface with a Knudsen number of 0.01 shifts downstream as the curvature is altered. The magnitude of this shift increases with the absolute value of the curvature. The aforementioned analysis of the spatial distribution of various flow mechanisms in the micro-nozzle underscores the complexity of flow states, ranging from continuum flow to free molecular flow, and highlights that the distribution positions of these flow states also change.

Changing the curvature of the needle valve cone has a significant impact on the distribution of Kn numbers in the nozzle. Through altering the curvature, the high-pressure region is pushed downstream along the microchannel, resulting in an increase in the transition flow area within the nozzle and a reduction in the overall Kn value. Specifically, under a 10% opening, the slip-flow zone occupies a significant portion of the microchannel. Modifying the curvature value of the needle valve causes the high-velocity zone to advance downstream along the microchannel, thereby increasing the proportion of the continuous flow zone and reducing the extent of the slip-flow zone. In the expansion section, increasing the curvature value of the needle valve expands the range of the transition zone, extending towards the nozzle outlet, and further narrowing the extent of the free molecular flow zone.

Figure 9 illustrates the flow mechanisms and the distribution of local Kn numbers in the micro-nozzle when the needle valve is opened at 50%. Comparing it with the figure at 10% opening, it is evident that as the opening increases, the proportion of the continuous flow area within the microchannel further expands, dominating the majority of the microchannel. However, the expansion section is predominantly occupied by the transition flow region, while the presence of the free molecular flow region becomes almost negligible. Moreover, increasing the curvature of the needle valve intensifies this tendency.

### 3.3. Multi-Scale Rarefied Flow Characteristics

Figure 10 illustrates the velocity distribution within the nozzle at a 90% opening after 1 ms. Notably, the acceleration of nitrogen molecules occurs in the vicinity of the microchannel outlet. As the curvature of the needle valve increases, there is a greater presence of high-speed fluid in the central region of the expansion section, resulting in a more uniform velocity distribution field with increased statistical representation of molecular samples. Consequently, in both the throat and expansion section, an augmented curvature of the needle valve leads to an enhancement in fluid velocity. Moreover, the magnitude of this effect is directly proportional to the absolute value of the curvature. The findings indicate that as high-pressure nitrogen passes through the valve tube and flows across the spool, its velocity progressively increases, with the highest velocity observed in the expansion section of the nozzle. This phenomenon arises due to the rapid variation in flow area, resulting in a significant pressure drop and steep gradient, which contributes to the attainment of maximum velocity at this location. Altering the curvature of the inclined surface of the spool causes a shift in the position of maximum velocity towards the outlet, accompanied by a gradual increase in its value. This displacement occurs because an increase in curvature value leads to an expansion of the flow area at the spool, albeit less dramatically than when the valve opening is small, resulting in a more stable velocity profile.

Through comparing different spool shapes, it is evident that the conical spool shape mitigates the velocity gradient in the spool region and the high-speed region near the outlet pipe connection, especially under large valve openings. Additionally, as the spool curvature increases, the velocity in the expansion section also increases. Consequently, a valve equipped with an arc-shaped spool yields the highest nitrogen flow rate at the nozzle when fully opened, thereby enhancing thrust performance.

### 3.4. Correlation of Flow Parameters with Curvature and Opening of Needle Valves

Figure 11 illustrates the instantaneous spatial distribution of DSMC sample particles (t = 1 ms) under different curvatures when the needle valve is opened at 10%, 50%, and 90%. The flow state in the micro-nozzle reaches a quasi-stable state by 1 ms. The number of molecules in the valve cavity and microchannel is relatively high, whereas the number of molecular samples in the expansion section is lower. At small valve openings, molecules tend to concentrate in the upper laryngeal cavity and microchannel due to the constriction effect of the throat. Specifically, at a 10% opening, the density of molecular samples is lower in the downstream one-third-length region of the microchannel. On the other hand, when the needle valve is fully opened at 90%, the molecular number density distribution in the micro-nozzle becomes more uniform.

Since the outlet is in a vacuum state, molecules disperse rapidly upon entering the nozzle’s expansion section from the microchannel, resulting in macroscopic fluid expansion. This expansion becomes more pronounced as the absolute value of the curvature increases. A deeper understanding of the relationship between flow parameters and the curvature and opening of needle valves can offer valuable insights for optimizing their structure [43,44]. Figure 12 shows the relationship between the curvature of the valve needle, valve opening, and thrust. Obviously, the curvature of the valve needle significantly affects the thrust. Increasing or decreasing the curvature can increase the thrust. The needle valve is designed with a positive curvature to increase the thrust more effectively.

## 4. Conclusions

This paper utilizes the DSMC method to analyze the flow behavior in a cold gas micro-thruster, specifically focusing on a micro-nozzle. The primary objective is to investigate the impact of the needle valve curvature on the flow characteristics. The study examines various aspects, including the spatial distribution, Kn number distribution, slip velocity distribution, and pressure distribution within the micro-nozzle. The key findings are summarized as follows:The free molecular flow is concentrated near the wall of the expansion section of the nozzle and in proximity to the nozzle outlet. The extent of this region diminishes as the opening of the nozzle increases. Moreover, altering the curvature of the needle valve cone leads to a reduction in the Kn number distribution within the nozzle, consequently decreasing the size of the free molecular flow zone.In the throat and expansion section, increasing the curvature of the needle valve positively influences fluid velocity, with a higher absolute curvature value resulting in higher fluid velocities. Comparing different spool shapes, the conical spool shape reduces the velocity gradient in the spool area and the high-speed region near the outlet pipe when the opening is large. As the curvature of the spool increases, the velocity in the expansion section also increases. Consequently, an arc-shaped spool valve achieves the highest nitrogen flow rate at the nozzle during wide openings, thereby enhancing thrust.When the needle valve is at a small opening, the molecules tend to concentrate in the upper laryngeal cavity and microchannel due to the constriction effect of the throat. As the outlet of the nozzle is under vacuum conditions, the molecules disperse upon entering the expansion section of the nozzle from the microchannel. The macroscopic behavior observed is fluid expansion. This expansion becomes more pronounced as the absolute value of the curvature increases.

## Figures and Tables

**Figure 1 micromachines-14-01585-f001:**
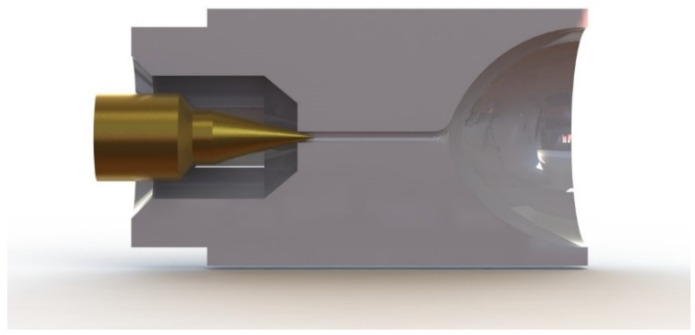
The micro-nozzle profile in the present study.

**Figure 2 micromachines-14-01585-f002:**
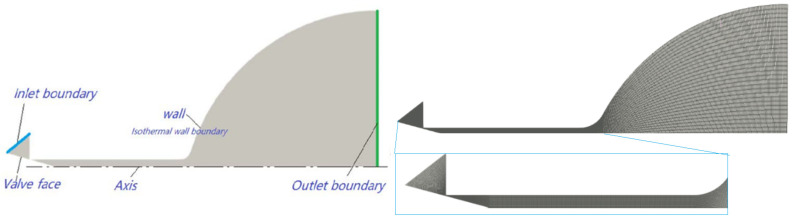
Computational domain and grids.

**Figure 3 micromachines-14-01585-f003:**
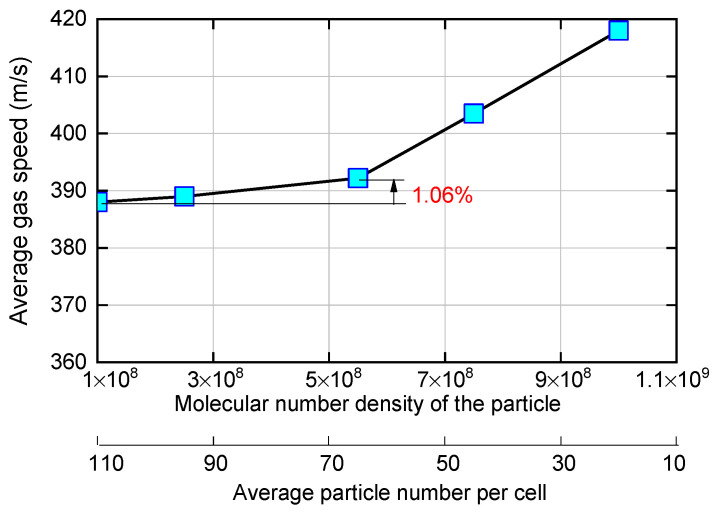
Effects of the molecular number density and average particle number per cell on the calculated average nitrogen gas speed at the outlet.

**Figure 4 micromachines-14-01585-f004:**
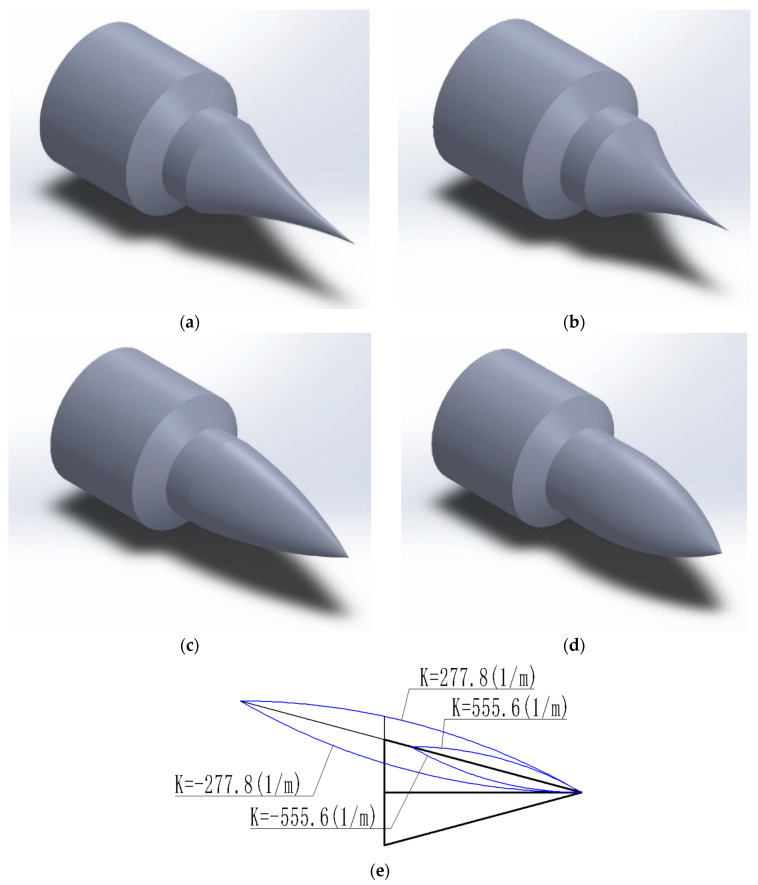
The three-dimensional structure of the valve needle with different curvatures: (**a**) K = −277.8 (1/m); (**b**) K = −555.6 (1/m); (**c**) K = 277.8 (1/m); (**d**) K = 555.6 (1/m); (**e**) curvatures of the valve needle.

**Figure 5 micromachines-14-01585-f005:**
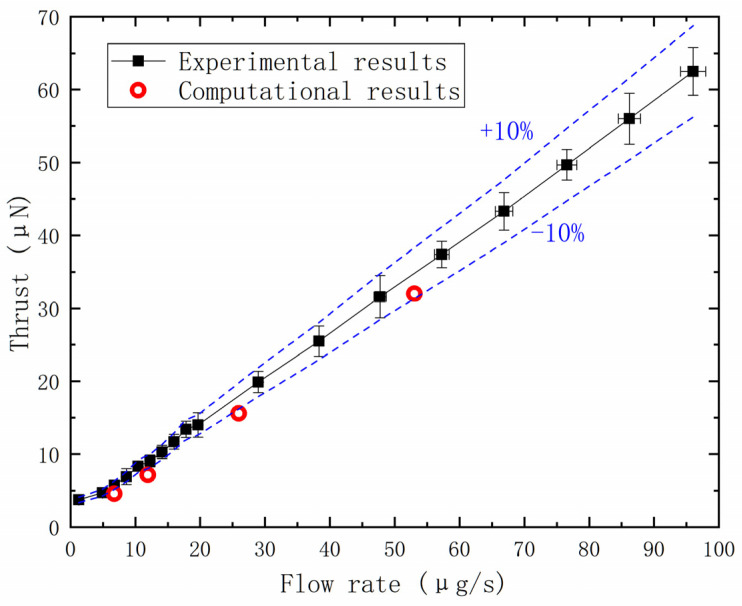
Comparison of the flow rate between the experimental and the computational results.

**Figure 6 micromachines-14-01585-f006:**
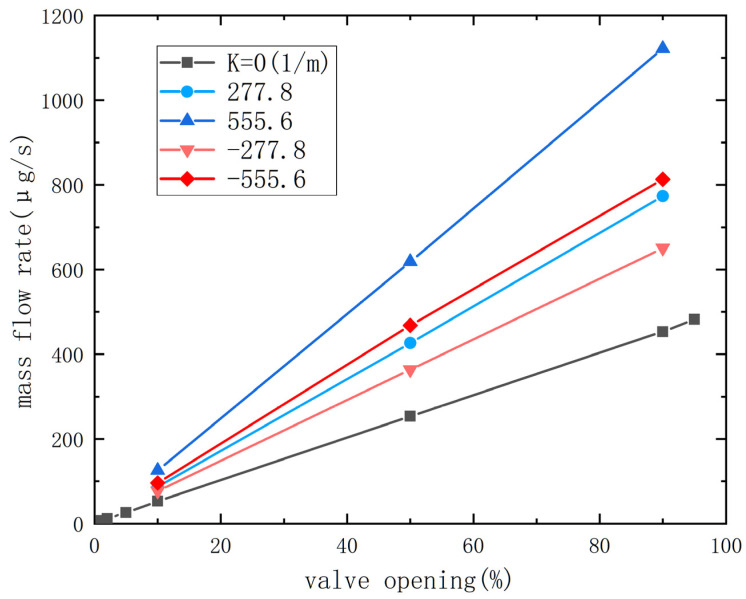
The relationship between mass flow rate and valve opening under different curvatures of the needle valve.

**Figure 7 micromachines-14-01585-f007:**
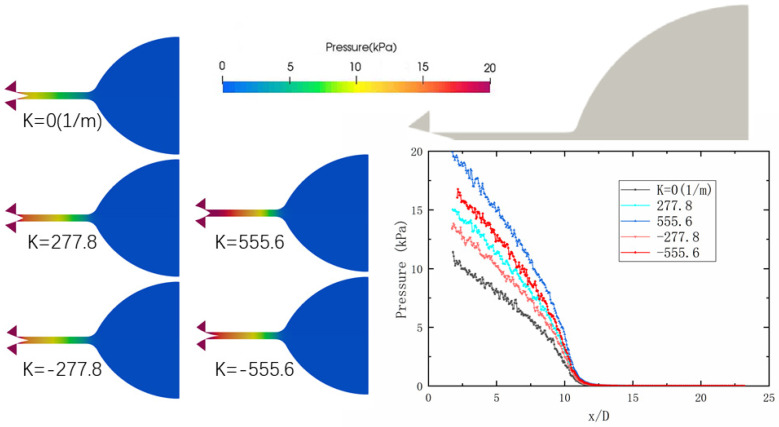
The pressure distribution map of each area in the nozzle at 50% opening at 1 ms.

**Figure 8 micromachines-14-01585-f008:**
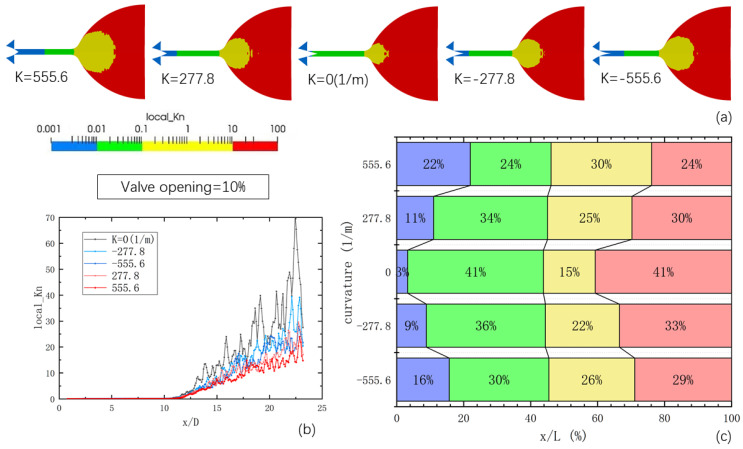
The local Kn number distributions of the internal area of the micro-nozzle at a 10% opening of the needle valve: (**a**) the distribution results for different curvatures with a 10% opening of the needle valve; (**b**) the variation curve of local Knudsen numbers along the axis with the length-diameter ratio for different curvatures; (**c**) the proportion of each region along the axis for different curvatures.

**Figure 9 micromachines-14-01585-f009:**
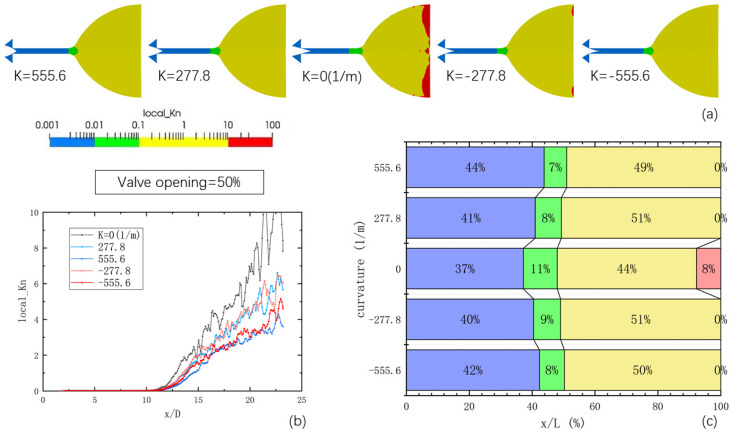
The local Kn number distribution results of the internal area of the micro-nozzle at a 50% opening of the needle valve: (**a**) the distribution results for different curvatures with a 50% opening of the needle valve; (**b**) the variation curve of local Knudsen numbers along the axis with the length-diameter ratio for different curvatures; (**c**) the proportion of each region along the axis for different curvatures.

**Figure 10 micromachines-14-01585-f010:**
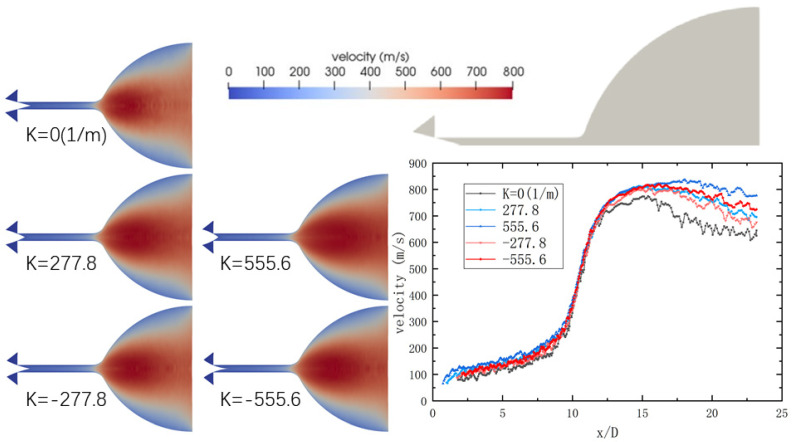
Velocity distributions in the nozzle at a 90% opening of the needle valve.

**Figure 11 micromachines-14-01585-f011:**
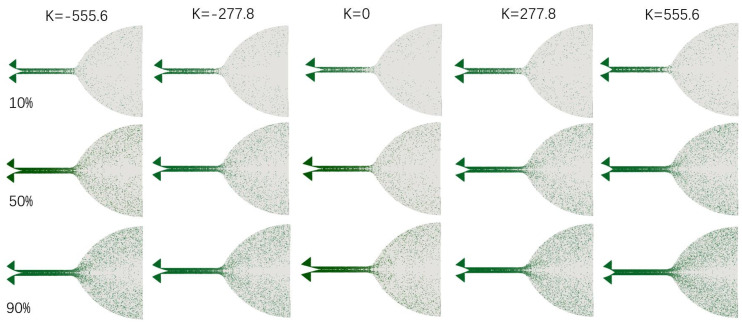
Particle position distributions in the axial plane.

**Figure 12 micromachines-14-01585-f012:**
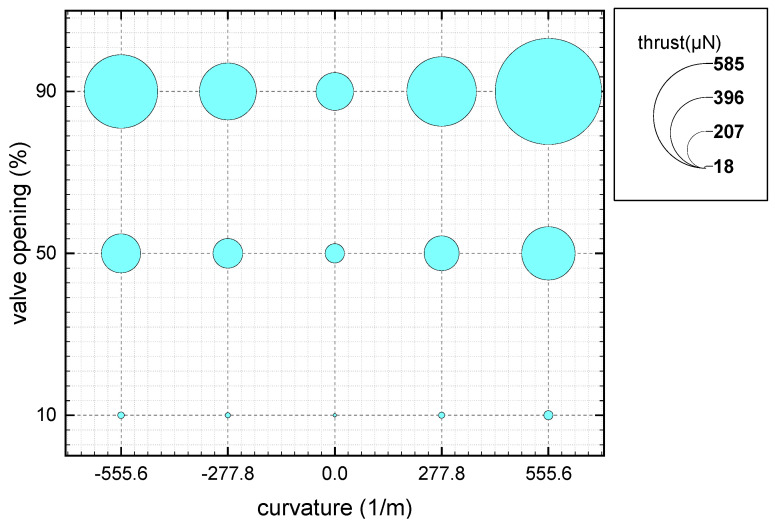
Bubble diagram of the thrust under different curvatures and different valve openings.

**Table 1 micromachines-14-01585-t001:** Effects of grids on the mean molecular velocity at the nozzle exit.

	Number of Axial Meshes	Number of Radial Meshes	Number of Circumferential Meshes	An Average Speed at the Exit (m/s)	Relative Error (%) (Grid7 as a Reference)
Grid1	30, 40, 120, 60	20	4	413.7	7.63
Grid2	30, 60, 140, 80	20	2	402.3	3.41
Grid3	30, 60, 140, 80	40	2	392.2	1.79
Grid4	30, 60, 140, 80	60	2	390.5	1.35
Grid5	50, 80, 160, 100	40	2	390.1	1.25
Grid6	70, 100, 180, 120	40	4	387.6	0.60
Grid7	70, 120, 200, 150	40	4	385.3	0.00

**Table 2 micromachines-14-01585-t002:** The percentage of mass flow rate relative to the cone (K = 0) under different curvatures.

Valve Opening (%)	K = 277.8 (1/m)	K = 555.6	K = −277.8	K = −555.6
10	164.73%	237.00%	145.32%	180.53%
50	168.11%	244.08%	143.42%	184.69%
90	170.54%	247.49%	143.51%	179.43%

## Data Availability

The data generated in this study are available on reasonable request from the corresponding author.

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
