# Peer review of "Direct Simulation Monte Carlo Simulation of the Effect of Needle Valve Structures on the Rarefied Flow of Cold Gas Thrusters"

_micromachines, 2023, doi:10.3390/mi14081585_

Round 1

Reviewer 1 Report

The referee report is attached.

The referee report is attached.

Reviewer 2 Report

I read the paper: DSMC simulation of the effect of needle valve structures on 2 the rarefied flow of cold gas thrusters. The paper is generally fine but needs the following amendments before publication:

1- The literature survey of the paper could be enhanced by noting the following DSMC papers when introducing DSCM:

Amiri, A., Roohi, E.*, Niazmand, H., Stefanov, S., "DSMC Simulation of Low Knudsen Micro/Nano Flows using Small Number of Particles per Cells," Journal of Heat Transfer, Vol. 135(10), 101008, 2013.

 Roohi, E.*, Darbandi, M., "Recommendations on Performance of Parallel DSMC Algorithm in Solving Subsonic Nanoflows," Applied Mathematical Modelling, Vol. 36 (5), pp. 2314-2321, 2012.

2- Once noting DSMC collision models, please introduce novel collision models such as the Simplified Bernoulli Trial (SBT), Generalized Bernoulli Trial (GBT), Symmetrized and Simplified Bernoulli Trial (SSBT), and hybrid collision method.

3- The authors should report the Particle per Cell (PPC) study effect as well.

4- Which Software did the author use? dsmcFaom? Please mention clearly with appropriate reference.

5- Figs. 7 and 8, please label each frame and report what is presented by each frame.

6- Fig. 10 is not informative and should be removed.

7- The quality of Fig. 11 should be improved, please improt in WMF format rather than TIF.

Round 2

Reviewer 1 Report

The referee report is attached.

Please re-read the manuscript for missing spaces and typos.

Reviewer 2 Report

The paper is fine; just in citing the papers, there were some errors, i.e.: do not mention the first name of the authors, i.e., Masoud Drabandi should be Darbandi and Roohi. Ehsan Roohi et al should be Roohi et al, E. Taheri et al should be Taheri et al., and similarly.

Author Response

Thank you for your comments on this article. It has been corrected.

Round 3

Reviewer 1 Report

-